# Influence of Surface Treatment and Accelerated Ageing on Biaxial Flexural Strength and Hardness of Zirconia

**DOI:** 10.3390/ma16030910

**Published:** 2023-01-18

**Authors:** Nadja Rohr, Angela Jacqueline Schönenberger, Jens Fischer

**Affiliations:** Biomaterials and Technology, Department of Reconstructive Dentistry, University Center for Dental Medicine Basel UZB, University of Basel, CH-4058 Basel, Switzerland

**Keywords:** zirconia implants, accelerated ageing, surface treatment, mechanical properties

## Abstract

The aim was to investigate how the surface treatment and the process of accelerated ageing of zirconia for dental implants affect the biaxial flexural strength and hardness. Zirconia discs with a diameter of 12.6 mm were subjected to either one of the following treatments: polishing (Zp); polishing and heat treatment at 1250 °C for 1 h (Zpt); machining (Zm); machining and heat treatment (Zmt); or sandblasting, acid-etching, and heat treatment (Z14) (n = 45 per group). Biaxial flexural strength and Martens hardness (HM) were measured without further treatment and after accelerated ageing for 5 h or 5 × 5 h according to ISO 13356 (n = 15 per group). Two-way ANOVA was applied to test the effect of surface treatment and ageing (α = 0.05). The reliability of the specimens was described with Weibull two-parameter distribution of biaxial flexural strength data. Overall, the surface treatment (*p* < 0.001) and ageing (*p* = 0.012) revealed a significant effect on biaxial flexural strength values, while HM was only affected by the surface treatment (*p* < 0.001) but not ageing (*p* = 0.160). Surface treatment significantly affected HM (*p* < 0.001) but not ageing (*p* = 0.160). The applied surface treatments affected the biaxial flexural strength and HM of zirconia. For accelerated ageing, a duration of both 5 h and 5 × 5 h is recommended to evaluate the effect of surface treatments. Zm was the most reliable surface as it was least affected by ageing and provided low standard deviations of biaxial flexural strength values.

## 1. Introduction

Zirconia is a suitable material for dental implants [1,2,3] with implant survival rates of 97.5% [4] to 98.5% after 3 years [5] and 94.3% after 5 years [6]. Zirconia is a high-strength oxide ceramic that displays a polycrystalline microstructure without a glass phase. Regarding the long-term success and stability of implants, osseointegration and consequently implant surface topography are of high relevance [7]. In order to achieve a fast osseointegration, the implant is postulated to require a rough surface [8,9], preferably a mean area roughness Sa of about 1.5 µm as suggested for titanium implants [10,11]. Different surface modifications have been applied to enhance the desired surface roughness for zirconia such as sandblasting, acid etching [11,12,13], laser modifications, sintering of ceramic slurry [14] or coating with porous layers of hydroxyapatite or calcium phosphate [1].

Zirconia occurs in three different polymorph crystal structures that are stable in different temperature ranges: monoclinic phase (room temperature up to 1170 °C), tetragonal phase (1170 °C–2370 °C with a decrease in volume of 5% (m → t)) and cubic phase (>2370 °C) [1,15]. To obtain the tetragonal structure at room temperature, metal oxides such as yttria (Y_2_O_3_), magnesia (MgO), limestone (CaO) or ceria (CeO_2_) can be added to zirconia [16,17,18]. These stabilizing metal oxides prevent the phase transition from tetragonal to monoclinic (t → m) and, therefore, stabilize the tetragonal phase [1,15]. Nevertheless, as a result of external stress such as sandblasting, phase transformation t → m still occurs [19]. The phase transformation leads to an increase in volume in the stress area preventing crack propagation which is referred to as transformation toughening. A heat treatment with temperatures higher than the phase transition can reverse the t → m transition and is, therefore, sometimes applied during implant production to increase ageing resistance [20]. The preferred zirconia ceramic for dental implants is 3Y-TZP, where zirconia is stabilized in the tetragonal phase by adding 3 mol% of Y_2_O_3_ [21]. However, the monoclinic fraction volume in 3Y-TZP is still around 10–15% [15]. This type of zirconia has a high compressive strength of 2000 MPa and a high flexural strength of 900–1200 MPa [22]. In presence of water the tetragonal phase is prone to t → m transformation, referred to as low temperature degradation (LTD) or hydrothermal ageing [21,23]. LTD can be simulated in vitro by subjecting zirconia to an autoclave treatment of 1 h at 134 °C at 2 bar, which corresponds to 98.7 kJ mol^−1^ and to an estimated ageing process of 1 year in vivo [21]. It is however unclear, if the surface treatment of zirconia affects its ageing and consequently the material strength.

Currently, on a dental implant different surfaces are favored, depending on the location: a rough surface in the endosseous part to achieve rapid osseointegration and a smooth surface on the transmucosal part for better cleanability. In general, the transmucosal part is polished. However, the roughness of an as-sintered (machined) 3Y-TZP surface is close to that of a polished surface [24] and may, therefore, also be considered for technical simplification. Additionally, some manufacturers apply a heat treatment after the surface structuring process to enhance the tetragonal phase and consequently increase the resistance to aging [20]. This heat-treatment process can change the surface topography of polished surfaces due to initiated grain growth, while for as-sintered surfaces, no changes in topography are expected [24,25]. In previous in vitro investigations, the cell viability and cell spreading of human-derived osteoblasts [24] as well as fibroblasts [25] were assessed on five different surfaces: an as-sintered surface and a polished surface, both with or without a thermal treatment to retrieve the tetragonal phase as well as a rough surface generated by etching, sandblasting and subsequent thermal treatment [11] as used in a commercially available zirconia implant. Those four experimental surfaces would be an option to simplify the elaborate production process of zirconia implants. All five surfaces provided an appropriate substrate for the cells and may be considered as viable implant surfaces. However, the surface treatment itself and the influence of the oral environment on the respective surface might affect the long-term stability of zirconia implants. The aim of the present study was, therefore, to investigate the effect of different surface treatments and accelerated ageing on the biaxial flexural strength and Martens hardness of zirconia that is used for implants.

## 2. Materials and Methods

Zirconia (3Y-TZP) discs with a diameter of 12.6 mm and a thickness of 1.9 mm were produced. The specimens were sliced over-dimensioned in the green state and sintered by the manufacturer (CeramTec, Plochingen, Germany). According to the manufacturer, the material was composed of 93.0 wt% ZrO2, 5.0 wt% Y_2_O_3_, 0.1 wt% Al_2_O_3_ and 1.9 wt% HfO_2_ with a grain size of 0.3 µm [25]. The discs were divided into 5 groups (n = 45 per group) and a surface treatment according to Table 1 was applied. The polishing procedure applied for Zp and Zpt was accomplished using diamond discs of nominal grit size 90, 70, 30, 15, 9, 3 and 1 µm with a speed of 800 rpm under constant water cooling. Heat treatment was applied for Zpt and Zmt to retrieve the tetragonal phase of zirconia. Specimens were, therefore, placed in a sintering furnace (VITA Zyrcomat 6100 MS, VITA, Bad-Säckingen, Germany) for 1 h at a temperature of 1250 °C. A cooling rate of 1.25 K/min was chosen. At a temperature of 400 °C, the furnace was opened. Z14 specimens were provided with a surface, which was identical to the surface of the endosseous part of a zirconia implant where clinical evidence is available [6] (ceramic.implant, VITA). That treatment was performed by the manufacturer by sandblasting the specimens for 20 s at a pressure of 5 MPa with 105 µm Al_2_O_3_ (Kavo EWL, Type 5423, Biberach, Germany) while specimens were placed at a 15 mm distance between the blasting tip and the surface of the sample. Afterwards, etching with hydrofluoric acid (38%) was conducted and specimens were heat-treated for 1 h at 1250 °C as described above.

Accelerated ageing according to ISO 13356 [26] was performed with 30 specimens per group. Specimens were, therefore, placed in an autoclave (Systec VX-150, Systec GmbH, Linden, Germany) and exposed to steam at 134 ± 2 °C for 5 h holding time under a pressure of 0.2 to 0.3 MPa. This procedure was performed once (5 h) or 5 times (5 × 5 h) with 15 specimens per group. No ageing was performed with the remaining 15 discs per group (baseline).

### 2.1. Biaxial Flexural Strength

Biaxial flexural strength was measured using the biaxial flexural test (piston-on-three-balls test, Z020 Zwick/Roell, Ulm, Germany) according to ISO 6872 [27]. Thirty specimens per group are recommended in ISO 6872. However, result analysis of a pre-test considering a power of 0.8 and a level of significance of 0.05 revealed that a number of specimens of 15 could be considered and were, therefore, chosen for this study. Diameter and height of the discs were measured with a digital calliper (Tesa technology, Tesa-Cal IP 67). Testing was performed with a crosshead speed of 1.5 mm/min at room temperature. Specimens were loaded with the non-treated side facing to the piston. The maximum centric tensile stress (σ) was calculated using the following equation and the number of fragments for each specimen was counted after the test.
σ = −0.2387 P(X − Y)/b2
where
σ is the maximum center tensile stress, in MPa;P is the total load causing fracture, in newtons;X = (1 + v) ln(r2/r3)2 + [(1 − v)/2](r2/r3)2;Y = (1 + v)[1 + ln(r1/r3)2] + (1 − v)(r1/r3)2.
where
v is Poisson’s ratio (0.25);r1 is the radius of the support circle, in mm;r2 is the radius of the loaded area, in mm;r3 is the radius of the specimen, in mm;b is the specimen thickness at the fracture origin, in mm.


### 2.2. Hardness

A hardness-testing machine (ZHU 2.5 Zwick/Roell GmbH, Ulm, Germany) with a Vickers intender was used for Martens hardness (HM) measurements. Examinations were performed according to ISO 14577 [28,29]. Measurements were conducted on the largest fracture piece of each sample from each of the 15 groups on the treated side at 2 different locations (n = 30 indentations per group). All specimen surfaces were loaded with a maximum load of 19.61 N (HV 2) and a crosshead speed of 5 N/s.

### 2.3. Scanning Electron Microscopy

Scanning electron microscopy (SEM) (XL 30 ESEM, Philips, Eindhoven, The Netherlands) was used to visualize microstructure and topography of the surfaces for baseline and aged (5 h, 5 × 5 h) specimens. Additionally, fracture analysis was conducted for each group of 2 selected fracture pieces. Before SEM recordings, the specimens were sputtered with a gold layer (BalTec SCD 005 Sputter Coating Unit, BAL-TEC Pfäffikon, Switzerland). Imaging was performed at an acceleration of 15 kV and magnifications of 5000× and 10,000×.

### 2.4. Statistical Analysis

Data were tested for normal distribution using Shapiro–Wilk test. Two-way ANOVA was applied to data of biaxial flexural strength, and HM to analyse the effects of surface treatment and ageing followed by post hoc Fisher LSD test (α = 0.05). In order to describe reliability of the specimens, Weibull two-parameter distribution was applied to biaxial flexural strength data according to ISO 6872 [27]. Weibull modulus and Weibull characteristic strength were calculated.

## 3. Results

### 3.1. Biaxial Flexural Strength

The mean biaxial flexural strength values of the specimens with the different surface treatments and ageing with post hoc comparisons are displayed in Table 2.

Overall, the surface treatment (*p* < 0.001) and ageing (*p* = 0.012) revealed a significant effect on biaxial flexural strength values. Significantly higher values were found for Zp (baseline: 1268 ± 263 MPa, 5 h: 1093 ± 53 MPa 5 × 5 h:1251 ± 274 MPa) than for all other groups (*p* < 0.001). Z14 tended to display the lowest values (baseline: 960 ± 64 MPa, 5 h: 1010 ± 93 MPa 5 × 5 h:1020 ± 59 MPa); however, the difference was only significant in comparison with Zp (*p* < 0.001) and Zpt (*p* = 0.022). No statistical difference was observed between Zpt, Zm and Zmt (*p* > 0.05).

At baseline, a mean biaxial flexural strength of 1075 ± 176 MPa of all groups was measured. Accelerated ageing for 5 h slightly decreased values (1030 ± 176 MPa) (*p* = 0.030), whereas ageing for 5 × 5 h slightly increased the values (1090 ± 173 MPa) (*p* = 0.496). However, only the values of the aged samples differed significantly from each other (*p* = 0.005). Weibull graphs for each ageing group are displayed in Figure 1. The Weibull characteristic strength for baseline was the highest for group Zp (1380 MPa) and the lowest for Z14 (987 MPa). After 5 h of accelerated ageing, the highest value was also shown for group Zp (1117 MPa) and the lowest for group Zm (1031 MPa), while after 5 × 5 h, the strength was highest for Zp (1362 MPa) and lowest for group Zmt (1031 MPa). The Weibull modulus (m), describing the reliability of zirconia with different surface treatments after the respective ageing periods, and the Weibull characteristic strength (σ) are shown in Table 2. The values of the groups Zp baseline, Zmt baseline, Zp 5 × 5 h, and Zpt 5 × 5 h did not add up to a linear fit. Weibull modulus (m) and Weibull characteristic strength (σ) were still calculated for those groups but are added in parentheses to Table 2.

The number of fracture pieces in correlation to the biaxial flexural strength is shown in Figure 2. Below about 1200 MPa, the specimens tended to fracture into 2–3 pieces and above 1200 MPa into 3–4 pieces.

### 3.2. Hardness

The HM of samples according to different surface treatments with ageing with post hoc comparisons is shown in Table 2. Overall, the surface treatment significantly affected the HM (*p* < 0.001) but not ageing (*p* = 0.160). The HM values overall were, therefore, pooled and were significantly highest for Zm (6424 ± 1745 N/mm^2^) and Zpt (6408 ± 1843 N/mm^2^) which did not differ from each other significantly (*p* = 0.948). The values for Z14 (5482 ± 1579 N/mm^2^) were significantly lower than all groups (*p* < 0.05) except Zp (5733 ± 1853 N/mm^2^) (*p* = 0.336).

### 3.3. Scanning Electron Microscopy

Scanning-electron-microscopy images of surface structure and fracture sites are shown in Figure 3. Grooves due to the polishing procedure can be observed for Zp and Zpt. Heat treatment of Zpt generated grain growth on the surface. Pores at the grain boundaries due to insufficiently sintered zirconia granules were visible for Zm and Zmt. A highly structured surface was visible for Z14. No difference in the microstructure was detected for all groups between the baseline and aged samples; therefore, only baseline samples are shown in Figure 3. All samples fractured from the highest tension point on the structured surface that was placed facing downwards in the specimen holder. Insufficiently melted zirconia granules were also visible at the fracture interfaces. The tension build-up in the fractured area may have caused the development of flaws at the boundaries of the granules. The fracture interfaces of all groups appeared similar. For Z14, the acid seemed to have penetrated the zirconia to a depth of around 20 µm, further expanding the space between granules.

## 4. Discussion

The aim of the present study was to analyze the influence of different surface treatments and accelerated ageing on the biaxial flexural strength and HM of 3Y-TZP. The results of this study show that surface treatments and accelerated ageing do have an impact on biaxial flexural strength, while HM is only affected by the surface treatment. This implies that surface treatments can affect the long-term stability of zirconia implants. The accelerated ageing of zirconia only affected biaxial flexural strength, possibly due to phase-transforming effects that are not detectable using hardness measurements and scanning electron microscopy. That ageing does not affect HM of 3Y-TZP confirms the findings of a previous study [30].

The polishing of 3Y-TZP reduces surface defects but also induces mechanical stress into the surface. Hence, both factors increased the biaxial flexural strength for group Zp even though the monoclinic portion has been reported to decrease [24]. However, the process of polishing induces mechanical stress and, therefore, leads to rather high and unpredictable results which do not line up as a linear regression within the Weibull statistics. The HM of Zp possibly decreased as plastic deformation may be easier with the observable polishing grooves or due to the lower monoclinic-phase content (Figure 3). The heat treatment of the polished surface (Zpt) relaxed the induced mechanical stress on the surface and the biaxial flexural strength consequently decreased. Heat treatment of 1250 °C is able to change the crystal structure and increases the tetragonal portion [24,25]. Additionally, a change in surface topography with observable grain growth was visible that may be responsible for the increase in HM due to the more homogenous surface. The surface topography of the selected samples was inspected using SEM after a flexural strength test and no changes in the topography were noted compared to previously investigated specimens [24,25]. As the hardness testing was conducted at least 5 mm apart from the fracture areas due to the positioning of the measuring device, it can be assumed that the HM values were not influenced by the flexural strength testing.

When micro-structuring the surface using sandblasting and hydrofluoric acid etching followed by heat treatment (Z14), the biaxial flexural strength as well as hardness decreased due to the surface defects that were created with the increase in the surface roughness. The sandblasting procedure is reported to increase the monoclinic phase, while hydrofluoric acid etching for 1 h with (38–40%) did not influence the crystal structure [11]. Heat treatment that was applied to enhance the ageing resistance decreased the monoclinic fraction [11].

Accelerated ageing in an autoclave for 5 h at 134 °C at 2 bar is reported to simulate in vivo ageing of 5 years [21]. With the applied pressure of 2 bar, monoclinic-phase transformation is reported to be induced and low-temperature degradation may occur due to the steam [21]. For the polished specimens Zp, the initial ageing for 5 h must have relieved the mechanically induced surface stress, resulting in a reduction in biaxial flexural strength. With prolonged ageing for 5 × 5 h, the effect of transformation toughening with the increased monoclinic portion must have resulted in the increase in strength. The same effect was observable for Zpt after 5 × 5 h of ageing. For both groups, prolonged ageing tended to decrease the HM. Similar findings were observed with polished specimens that were autoclaved at 2 bar at 134 °C for 1 or 10 h, respectively [31].

With the machined samples, the standard deviation of biaxial flexural strength was the lowest and the material was not affected by ageing, implying that this is the surface with the highest predictability for clinical use. With the applied heat treatment for Zmt, the biaxial flexural strength was increased initially, but the Weibull modulus could not be calculated due to the scattering of the values. Hence, heat treatment does not seem beneficial for machined surfaces regarding the mechanical properties. For Z14, the effect of ageing on the biaxial flexural strength increase was already observable after 5 h due to its increased surface area. With prolonged ageing, HM may have decreased due to the increased amount of monoclinic phase present on the surface.

Although the ISO 13356 [26] recommends an accelerated ageing duration of 5 h, it is recommended to also evaluate a longer period to further assess the effects of a surface treatment on zirconia. A correlation was found between the monoclinic ratio at baseline measured in a previous study [24] and the biaxial flexural strength at baseline obtained in the present investigation (Figure 4). For Zp, a maximum in biaxial flexural strength was found with a monoclinic ratio around 8%. Those results suggest that polishing has a stronger impact on biaxial flexural strength than the monoclinic ratio. Zp showed a strong increase in strength compared to the other groups. These findings of course are to be interpreted very carefully because different surfaces affect the results of X-ray diffraction in a different way. Nevertheless, assessing the monoclinic ratio after aging may provide further insights into phase-transformation mechanisms and the effect on mechanical behavior and should be performed in future investigations.

An almost linear connection between biaxial flexural strength and the number of fracture pieces was observed; hence, the higher the force necessary to fracture the specimens, the more fracture parts can be expected, suggesting that higher internal stress was present in the specimens. The critical value where three or more pieces can be expected was around 1200 MPa. From a mechanical point of view, Zm is an interesting option due to its resistance to aging and in regard to production costs. Further in vitro and in vivo studies must be performed to verify the present results. From a mechanical point of view, Zm is an interesting option due to its resistance to aging and in regard to production costs. Further in vitro and in vivo studies must be performed to verify the present results.

## 5. Conclusions

To interpret all results in detail and to develop and improve zirconia-implant surface treatments, further investigations would be needed such as measurement of monoclinic to tetragonal ratio. Within the limitations of this study, it can be stated that:Surface treatments affect the biaxial flexural strength and hardness of zirconia.Accelerated ageing in an autoclave at 134 °C at 2 bar for both 5 h and 5 × 5 h can be recommended to evaluate the effect of surface treatments of zirconia on biaxial flexural strength.A machined zirconia surface is least affected by ageing and provides low standard deviations of biaxial flexural strength values.

## Figures and Tables

**Figure 1 materials-16-00910-f001:**
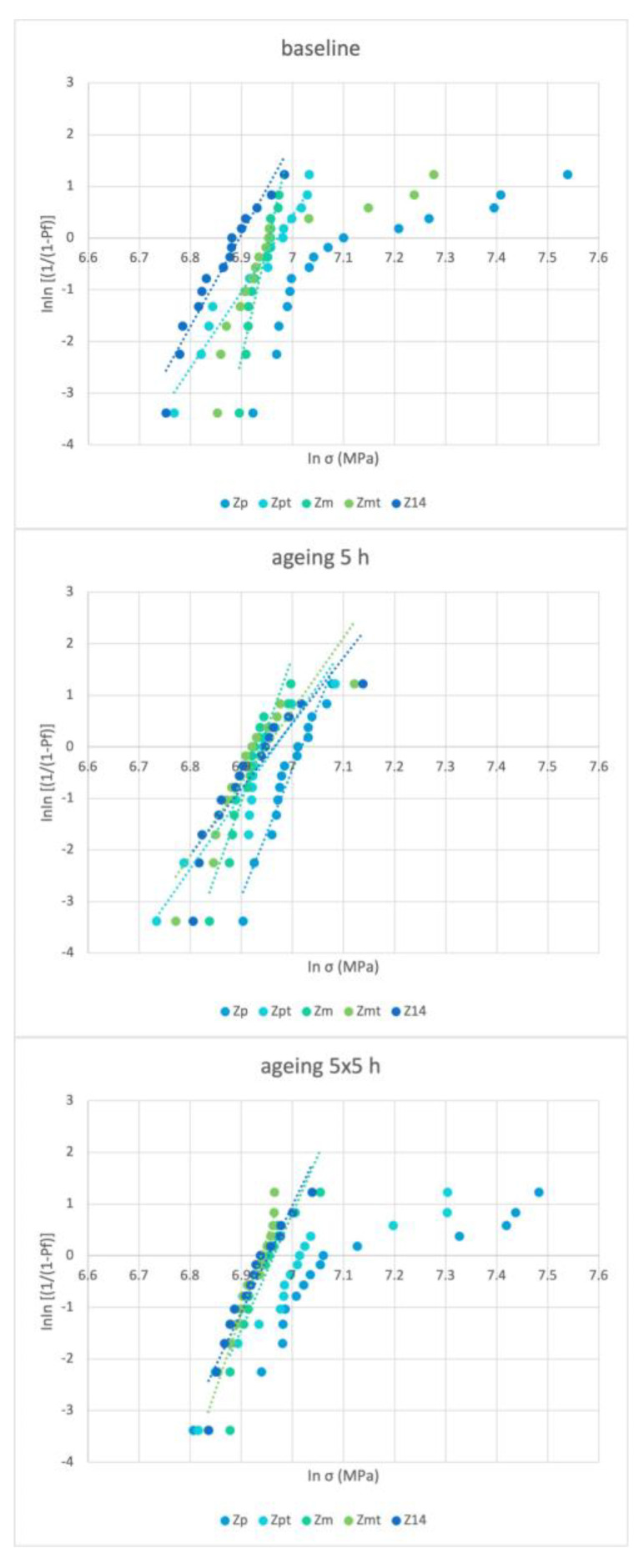
Weibull statistics of biaxial flexural strength values of zirconia with different surface treatments (Zp, Zpt, Zm, Zmt, and Z14) at baseline and after ageing 5 h and ageing 5 × 5 h.

**Figure 2 materials-16-00910-f002:**
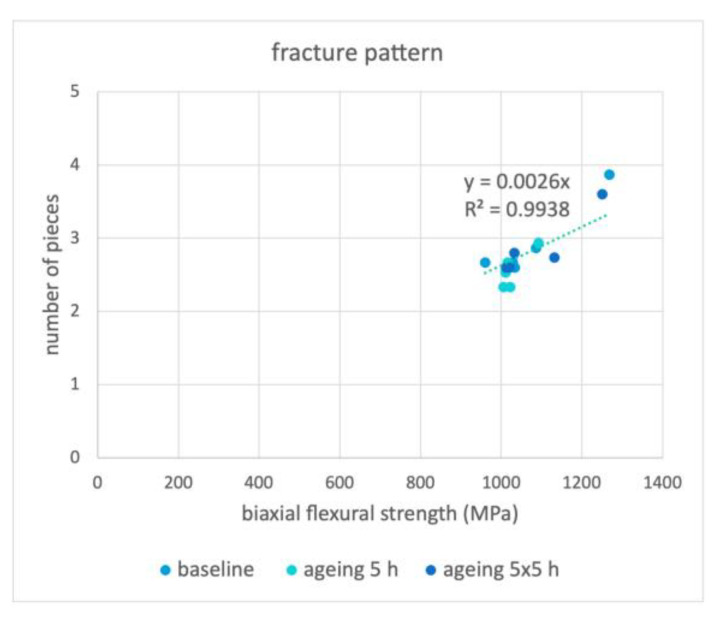
Mean number of fractured pieces correlated to the mean biaxial flexural strength of Zp, Zpt, Zm, Zmt and Z14 for the baseline, ageing 5 h, and ageing 5 × 5 h groups.

**Figure 3 materials-16-00910-f003:**
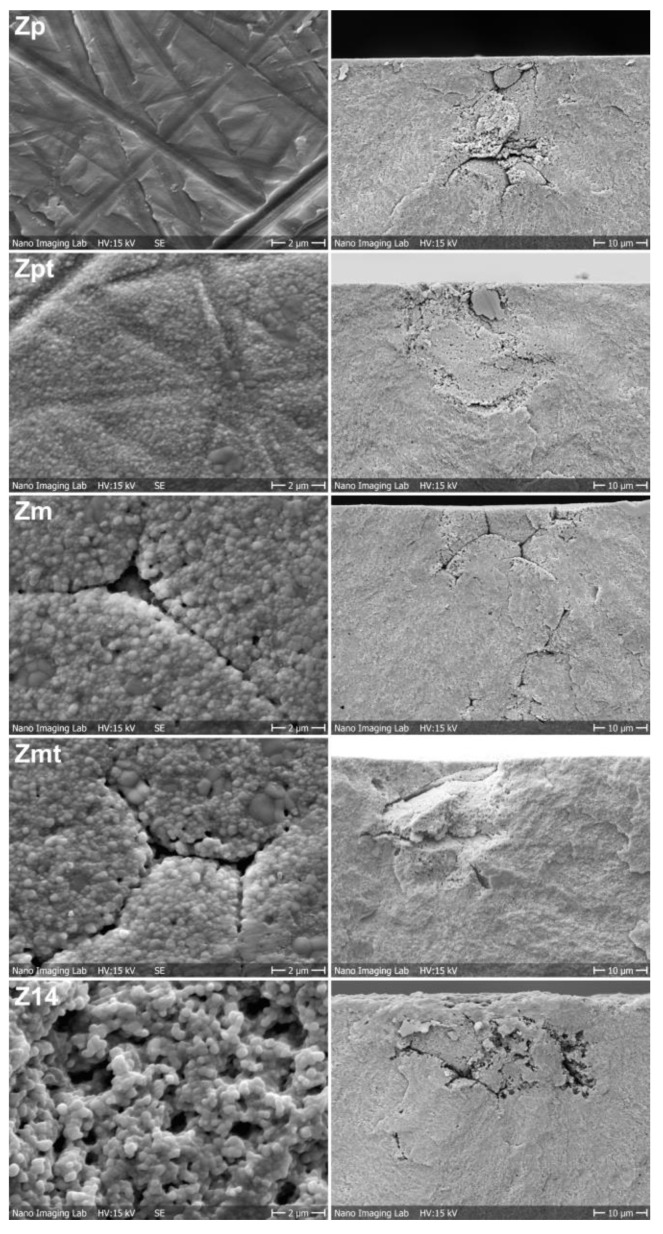
Scanning-electron-microscopy images of surface treatments of groups Zp, Zpt, Zm, Zmt and Z14 in left column. The fracture initiation site after conducting biaxial flexural strength testing for the respective group is displayed in the right column.

**Figure 4 materials-16-00910-f004:**
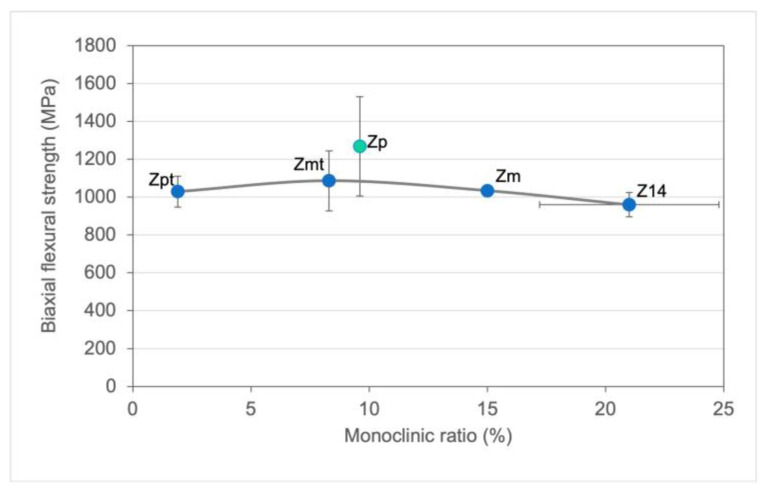
Biaxial flexural strength mean and standard deviations correlated to the monoclinic ratio of baseline samples that were obtained in a previous study [24].

**Table 1 materials-16-00910-t001:** Surface treatments of zirconia surfaces for the respective groups.

Code	Surface Treatments
Zp	Polished
Zpt	Polished, heat-treated 1 h 1250 °C
Zm	Machined (as-sintered)
Zmt	Machined, heat-treated 1 h 1250 °C
Z14	Sandblasted Al_2_O_3_ 105 µm, etched 1 h hydrofluoric acid 38–40%, heat-treated 1 h 1250 °C

**Table 2 materials-16-00910-t002:** Mechanical properties of zirconia with different surface treatments (Zp, Zpt, Zm, Zmt, and Z14). The values are presented using mean and standard deviations for biaxial flexural strength and Martens hardness (HM).

		Zp	Zpt	Zm	Zmt	Z14
Biaxial flexural strength (MPa)	Baseline	1268 ± 263 ^A,a^	1029 ± 82 ^B,C,a,b^	1034 ± 28 ^B,C,a^	1086 ± 158 ^B,a^	960 ± 64 ^C,a^
	Ageing 5 h	1093 ± 53 ^A,b^	1023 ± 81 ^A,a^	1011 ± 42 ^A,a^	1006 ± 83 ^A,a^	1016 ± 93 ^A,a^
	Ageing 5 × 5 h	1251 ± 274 ^A,a^	1132 ± 173 ^B,b^	1033 ± 52 ^B,C,a^	1012 ± 41 ^C,a^	1020 ± 59 ^C,a^
Weibull modulus/characteristic strength (MPa)	Baseline	(7.2)/(1380)	7.0/1065	7.0/1046	(7.1)/(1161)	6.9/987
	Ageing 5 h	7.2/1117	7.0/1060	6.9/1031	7.0/1043	7.0/1058
	Ageing 5 × 5 h	(7.2)/(1362)	(7.1)/(1206)	7.0/1058	6.9/1031	6.9/1046
HM (N/mm^2^)	Baseline	5500 ± 1415 ^A,a^	6758 ± 1400 ^B,a^	6682 ± 1348 ^B,a^	6185 ± 1671 ^A,B,a^	5769 ± 1247 ^A,B,a^
	Ageing 5 h	5754 ± 2032 ^A,a^	6650 ± 1810 ^A,a^	6243 ± 1813 ^A,a^	6117 ± 1758 ^A,a^	5815 ± 1523 ^A,a^
	Ageing 5 × 5 h	5611 ± 2443 ^A,a^	5816 ± 2153 ^A,B,a^	6352 ± 2032 ^A,a^	6108 ± 2042 ^A,a^	4869 ± 1776 ^B,a^

Statistical differences between groups (Fisher LSD test, *p* < 0.05) are indicated with differing superscript letters (uppercase horizontal comparison between groups; lowercase vertical comparison of ageing). Values of groups Zp, Zmt baseline and Zp, Zpt 5 × 5 h Zp did not add up to a linear fit. Weibull modulus (m) and Weibull characteristic strength (σ) were still calculated but added in parentheses.

## Data Availability

The data presented in this study are available on request from the corresponding author.

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
