# Peer review of "Influence of Surface Treatment and Accelerated Ageing on Biaxial Flexural Strength and Hardness of Zirconia"

_materials, 2023, doi:10.3390/ma16030910_

Round 1

Reviewer 1 Report

In General, this manuscript investigated an interesting and relevant topic. However, some aspects of the introduction, materials and methods, rational of the study must be reviewed before publication is possible.

Introduction

1.         No data information provided regarding to the issues of the study concerned (surface treatment/aging effect on zirconia). The author would like to study on the surface treatment; I think the main ones would be heat treatment and machining with or without other surface treatment methods. However, no information from previous studies about how heat treatment/machining methods and aging effect on zirconia (rational and mechanism) to allow readers to get the idea of research gap and research rational.

2.         What type of zirconia the author studied? If 3Y-TZP, please specify.

Materials and methods

3.         Sample size calculation should be provided.

4.         Line 183, give space between “h and did not”.

Author Response

In General, this manuscript investigated an interesting and relevant topic. However, some aspects of the introduction, materials and methods, rational of the study must be reviewed before publication is possible.

Response: We would like to thank you for the effort regarding the review of our manuscript. We have considered your recommendations and have tracked the amendments in the manuscript.

Introduction

  1. No data information provided regarding to the issues of the study concerned (surface treatment/aging effect on zirconia). The author would like to study on the surface treatment; I think the main ones would be heat treatment and machining with or without other surface treatment methods. However, no information from previous studies about how heat treatment/machining methods and aging effect on zirconia (rational and mechanism) to allow readers to get the idea of research gap and research rational.

Response: Thank you for your comment. We have rewritten parts of the last paragraph of the introduction line 66-78 to meet the request of the reviewer as follow: "In general, the transmucosal part is polished. However, the roughness of an as sintered (machined) 3Y-TZP surface is close to that of a polished surface [24] and may therefore also be considered for technical simplification. Additionally, some manufacturers apply a heat-treatment after the surface structuring process to enhance the tetragonal phase and consequently increase the resistance to aging [20]. This heat-treatment process can change the surface topography of polished surfaces due to initiated grain growth, while for as sintered surfaces, no changed in topography are expected [24, 25]. In previous in vitro investigations, the cell viability and cell spreading of human derived osteoblasts [24] as well as fibroblasts [25] was assessed on five different surfaces: an as sintered surface and a polished surface, both with or without a thermal treatment to retrieve the tetragonal phase as well as a rough surface generated by etching, sandblasting and subsequent thermal treatment [11] as used in a commercially available zirconia implant. Those four experimental surfaces would be an option to simplify the elaborate production process of zirconia implants All five surfaces provided an appropriate substrate for the cells and may be considered as viable implant surfaces. But the surface treatment itself and the influence of the oral environment on the respective surface might affect the long-term stability of zirconia implants. The aim of the present study was therefore to investigate the effect of different surface treatments and accelerated ageing on biaxial flexural strength and Martens hardness of zirconia that is used for implants."

  1. What type of zirconia the author studied? If 3Y-TZP, please specify.
    Response: Yes indeed 3Y-TZP was used and the detail was added to the introduction and at the beginning of MM. The exact composition is given in MM line 88-89: "93.0 wt% ZrO2, 5.0 wt% Y2O3, 0.1 wt% Al2O3 and 1.9 wt% HfO2 with a grain size of 0.3 µm."

Materials and methods

  1. Sample size calculation should be provided.
    Response: The reason for using the respective sample sizes was provided in MM line 120-123: Thirty specimens per group are recommended in ISO 6872. However, result analysis of a pre-test considering a power of 0.8 and a level of significance of 0.05 revealed, that a number of specimens of 15 can be considered and were therefore chosen for this study."

  1. Line 183, give space between “h and did not”.

Response: Thank you for noticing a space was added.

Reviewer 2 Report

Dear Respected author;

Thanks for preparing and presenting the manuscript entitled "Influence of surface treatment and accelerated ageing on biaxial 2 flexural strength and hardness of zirconia"

I found it  informative and worth reading, yet I have few queries for you;

1. What was your cutting tool that was used to cut the zirconia specimens into discs with diameter 12.6?

2. For Z14 group identical to Ref [6] 5 years  clinical trails; the exact implant treated protocol is not available in that reference, so where to find please?

3. Regarding the hardness test, what do you think about the effect of fracture stresses of flexural strength specimens on the validity of the hardness test results?

4. Regarding SEM graphs, Is the left column graphs having the same magnification?

I have noticed increased magnifications power between groups, while the ruler bar same length!

Author Response

Dear Respected author;

Thanks for preparing and presenting the manuscript entitled "Influence of surface treatment and accelerated ageing on biaxial 2 flexural strength and hardness of zirconia"

I found it  informative and worth reading, yet I have few queries for you;

Response: We would like to thank you for the effort regarding the review of our manuscript. We have considered your recommendations and have tracked the amendments in the manuscript.

  1. What was your cutting tool that was used to cut the zirconia specimens into discs with diameter 12.6?
    Response: The cutting was conducted by the manufacturer CeramTec and unfortunately, they did not provide this information.
  2. For Z14 group identical to Ref [6] 5 years  clinical trails; the exact implant treated protocol is not available in that reference, so where to find please?
    Response: That is correct, the reference was provided to refer to the clinical survival rate. The treatment of the implants was not described in this reference but is described in MM of the present study line 103-108: "That treatment was performed by the manufacturer by sandblasting the specimens for 20 s at a pressure of 5 MPa with 105 µm Al2O3 (Kavo EWL, Type 5423, Biberach, Germany) while specimens were placed at a 15 mm distance between the blasting tip and the surface of the sample. Afterwards, etching with hydrofluoric acid (38%) was conducted and specimens were heat-treated for 1 h at 1250°C as described above."
  3. Regarding the hardness test, what do you think about the effect of fracture stresses of flexural strength specimens on the validity of the hardness test results?
    Response: The surface topography of selected samples was inspected using SEM after flexural strength test and no changes in the topography was noted compared to previously investigated specimens [24, 25]. As the hardness testing was conducted at least 5 mm apart from the fracture areas due to the positioning of the measuring device, it can be assumed that HM values were not influenced by the flexural strength testing. Those two sentences were added to the Discussion line 259-264.
  4. Regarding SEM graphs, Is the left column graphs having the same magnification? I have noticed increased magnifications power between groups, while the ruler bar same length!
    Response: The magnification bars were added automatically by the SEM operating software and not changed with any image processing. We assume that they are correct as they are also coherent with the observations during our other studies with these surfaces [24, 25].